# Smartphone-Controlled Aptasensor for Voltammetric Detection of Patulin in Apple Juice

**DOI:** 10.3390/s24030754

**Published:** 2024-01-24

**Authors:** Arzum Erdem, Huseyin Senturk

**Affiliations:** Analytical Chemistry Department, Faculty of Pharmacy, Ege University, Bornova 35100, Izmir, Turkey

**Keywords:** patulin, aptasensor, apple juice, smartphone-controlled analysis

## Abstract

Patulin (PAT) is a mycotoxin that adversely affects the health of humans and animals. PAT can be particularly found in products such as apples and apple juice and can cause many health problems if consumed. Therefore, accurate and sensitive determination of PAT is very important for food quality and human and animal health. A voltammetric aptasensor was introduced in this study for PAT determination while measuring the changes at redox probe signal. The limit of detection (LOD) was found to be 0.18 pg/mL in the range of 1–10^4^ pg/mL of PAT in buffer medium under optimum experimental conditions. The selectivity of the PAT aptasensor against ochratoxin A, fumonisin B1 and deoxynivalenol mycotoxins was examined and it was found that the aptasensor was very selective to PAT. PAT determination was performed in an apple juice medium for the first time by using a smartphone-integrated portable device, and accordingly, an LOD of 0.47 pg/mL was achieved in diluted apple juice medium. A recovery range of 91.24–93.47% was obtained for PAT detection.

## 1. Introduction

Mycotoxins are toxic substances produced by certain types of fungi. They can grow on a variety of crops and food products, including grains, fruits, and vegetables. Mycotoxins can contaminate food at any stage of production, from growth to storage and processing, and can pose a significant threat to human and animal health. Some common types of mycotoxins include aflatoxins, ochratoxins, deoxynivalenol, fumonisins, and PAT. The toxicity of mycotoxins depends on the dose and duration of exposure, as well as individual susceptibility factors. Exposure to mycotoxins can cause a range of adverse health effects, including liver and kidney damage, immune system suppression, gastrointestinal irritation, and even cancer. Mycotoxins are a significant concern for food safety, and efforts have been made to reduce their levels in food products through good agricultural and manufacturing practices, as well as by regular testing and monitoring [1,2,3,4,5,6].

PAT is a mycotoxin produced by certain molds, such as *Penicillium*, *Aspergillus*, and *Byssochlamys* species. PAT is commonly found in apples and apple juices [7,8,9]. The level of PAT in food is regulated by many countries and food products containing PAT above the threshold limit are not sold. According to Commission Regulation (EC) No 1881/2006, the maximum limit of PAT in fruit juices, cider and other fermented beverages derived from apples or containing apple juice is set at 50 µg/kg [10].

The mycotoxin PAT can have detrimental effects on human health, with the severity of its impact depending on the dose and duration of exposure, as well as on individual susceptibility factors. PAT exposure at low levels is not typically considered a significant risk to human health, while high levels of exposure can result in a range of adverse health effects. These effects may include gastrointestinal irritation, resulting in symptoms such as nausea, vomiting, and diarrhea, as well as suppression of the immune system, which can increase an individual’s susceptibility to infections. Moreover, PAT is classified as a potential carcinogen, indicating its potential to cause cancer [11,12,13,14].

PAT can be determined by techniques such as chromatographic methods, ELISA and PCR [15,16,17]. While these techniques have some advantages, they also have disadvantages. The disadvantages of chromatographic techniques are that they are expensive, require specialized equipment, require a long sample preparation time and have slightly lower sensitivity. Although the selectivity of the ELISA method is good, the disadvantages are that cross-reactions can be observed and that it is a time-consuming technique. Although the sensitivity of the PCR method is good, the disadvantages are that it requires expert analysis and that false positive or false negative results can be observed depending on the primers.

There are several studies that present aptasensors specific to PAT [18,19,20,21,22,23,24,25,26,27,28,29]. In these studies, biosensors based on electrochemical [18,19,22,23,24,25], optical [20,27,28,29] and SERS-based [21] methods are introduced. In recent years, electrochemical methods have emerged as a promising alternative for the detection of different analytes. Recent years have seen a great deal of research and development on electrochemical sensors, with a focus on improving their resilience, usefulness, and appropriateness for new sensing goals. The number of various nanomaterials used in sensor design has risen, particularly with the advancement of nanotechnology, and the developed sensors’ performances have significantly improved. These nanomaterials can provide a large surface area and offer unique electronic properties that improve sensor performance. Furthermore, it is possible to detect biological analytes, such as proteins and different nucleic acids, with a high degree of specificity by integrating biological recognition components, such as enzymes, antibodies, DNA, aptamer etc. with electrochemical sensors. Miniaturized electrochemical sensor development is becoming more popular for applications such as those associated with continuous monitoring and point-of-care diagnostics. These sensors are appropriate for environmental monitoring and personalized healthcare because of their low cost, portability, and real-time measurement capabilities. In biosensor studies, a variety of electrochemical techniques are employed to appraise and analyze the performance of sensors for various applications. Among these techniques, cyclic voltammetry, differential pulse voltammetry, square wave voltammetry, amperometry and electrochemical impedance spectroscopy techniques are widely used. The use of electrochemical techniques offers several potential advantages, for example, high sensitivity, selective analysis, low detection limits, fast analysis, suitability for miniaturization, and low cost [30,31].

In this study, a voltammetric aptasensor specific to PAT was developed using a single-use electrode by measuring the changes at the signal of the redox probe in the absence/presence of PAT. In contrast to earlier studies [18,20,21,22,24,26,27,28,29], our aim is to obtain a lower limit of detection for PAT. To the best of our knowledge, this is the first aptasensor study in the literature presenting the determination of PAT by a smartphone-integrated portable device. The limit of detection for PAT was estimated and the selectivity of the aptasensor was tested against to the mycotoxin interferents: ochratoxin A (OTA), fumonisin B1 (FB1) and deoxynivalenol (DON). PAT determination was performed in apple juice medium using a smartphone-controlled aptasensor connected to a portable device. In this medium, the limit of determination was calculated, and a recovery study was performed for PAT determination.

## 2. Materials and Methods

### 2.1. Reagents and Materials

The PAT-specific aptamer sequence was chosen in accordance with the literature [18,32]. The PAT-specific aptamer sequence was synthesized and purchased from Ella Biotech (Fürstenfeldbruck, Germany).

The sequence of the PAT-specific aptamer is as follows:

5′-GGC CCG CCA ACC CGC ATC ATC TAC ACT GAT ATT TTA CCT T-NH_2_-3′

The stock solution of the PAT-specific DNA aptamer (APT) was prepared in fresh ultrapure water at a concentration of 1000 µg/mL and stored at −20 °C. The diluted solutions of aptamer (in the range of 0.25 µg/mL to 5 µg/mL) were prepared by using 5 mM Tris-HCl buffer (TBS, containing 20 mM NaCl).

PAT, fumonisin B1 (FB1) solution, deoxynivalenol (DON) solution, and ochratoxin A (OTA) solution were purchased from Sigma-Aldrich. The stock solution of PAT (i.e., 2500 µg/mL) was prepared in ethyl acetate and stored at −20 °C. The diluted solutions of PAT, FB1, DON, and OTA were prepared in phosphate buffer solution (PBS; 50 mM, pH 7.40). Safety information of the mycotoxins used in the study is given in the Appendix A. N-(3-dimethylaminopropyl)-N′-ethylcarbodiimide hydrochloride (EDC), N-hydroxysuccinimide (NHS), potassium chloride (KCl), sodium chloride (NaCl), acetic acid, potassiumdihydrogen phosphate, di-potassium hydrogen phosphate, potassium hexacyanoferrate (II) trihydrate, and potassium hexacyanoferrate (III) were purchased from Sigma-Aldrich (Steinheim, Germany).

All chemicals were of analytical reagent grade and supplied by global suppliers. Milli-Q^®^ (Merck, Istanbul, Turkey) ultra-pure water (resistivity 18.2 MΩ·cm @ 25 °C; total organic carbon (TOC) ≤ 5 ppb) was used to prepare all aqueous solutions freshly.

### 2.2. Apparatus

Electrochemical measurements were carried out by using µAUTOLAB III electrochemical analysis system with NOVA 1.11.1 software (Eco Chemie, Utrecht, The Netherlands). Differential pulse voltammetry (DPV) measurements were performed in a Faraday cage (Eco Chemie, The Netherlands). All measurements were performed with a traditional three-electrode system using a disposable pencil graphite electrode (PGE), a platinum wire, and an Ag/AgCl/KCl/3 M (BAS, Model RE-5B, W. Lafayette, LA, USA) as working, counter, and reference electrode, respectively.

### 2.3. Procedure

The procedure of the developed aptasensor for PAT determination is given below.

a. Activation of the electrode surface: The PGE surface was first electrochemically activated for 30 s by applying a potential of +1.2 V in acetate buffer solution (500 mM ABS, pH 4.8). Chemical activation of the electrodes was then performed with 5 mM EDC/8 mM NHS [33] for 30 min. The electrode surface was activated with EDC/NHS for covalent binding of the amino-labelled aptamer sequence onto the electrode surface [34].

b. Immobilization of the aptamer on the electrode surface: PGE was immersed into the sample containing 2.5 µg/mL PAT-specific aptamer in 40 µL and immobilization was then performed for 60 min. The electrode was then washed with TBS for 3 s to eliminate non-specific binding.

c. Interaction of aptamer and PAT on the electrode surface: Different concentrations of PAT (1–10^5^ pg/mL) were prepared in PBS, and the electrode was then immersed into each of the PAT solutions. Aptamer and PAT interacted on the electrode surface for 30 min. The electrode was then washed with PBS for 3 s to eliminate non-specific binding.

The whole procedure was carried out at room temperature.

d. DPV measurements: DPV measurements were performed in 5 mM [Fe(CN)_6_]^3−/4−^ redox probe solution. A redox probe solution of 5 mM was prepared in PBS containing 100 mM KCl. Voltammetric transduction was performed by measuring the redox probe signal observed at +0.22 V.

### 2.4. Selectivity Studies

Selectivity studies were carried out to demonstrate the selectivity of aptasensor to its target, PAT. In this context, selectivity studies were carried out with 10 ng/mL OTA, DON and FB1 mycotoxins. In the selectivity studies, the same experimental procedure was followed as mentioned in the Section 2.3.

### 2.5. Application of PAT Aptasensor to Apple Juice Samples

In order to demonstrate the applicability of the developed aptasensor to real samples, PAT determination was performed in apple juice medium. Apple juice samples were purchased from a local market in Turkey. Apple juice samples were diluted 1:50 in PBS in accordance with the data in the literature [19]. Different concentrations of PAT were prepared in apple juice medium and the same procedure was followed. DPV measurements were then performed.

### 2.6. Applications of PAT Aptasensor with Smartphone-Integrated Portable Device

The application of the PAT aptasensor was examined by a smartphone-integrated portable device. DPV measurements were performed by using a Galvanoplot analyzer system with GalvanoPlot TOUCH v1.1.0 software (Solar Biotechnology, Izmir, Turkey). The portable electrochemical analyzer was connected with the smartphone via a USB Type-C port. The three-electrode system was set up and DPV measurements were performed using the GalvanoPlot TOUCH software installed on the smartphone.

## 3. Results and Discussion

In this study, a voltammetric transduction was employed for the determination of PAT. Differential pulse voltammetry (DPV) is a highly sensitive electrochemical technique widely used in biosensor studies because of the many benefits it provides. The most important advantages are its sensitivity, selectivity, its enabling of measurements over a wide potential range, low background current and fast measurement. For these reasons, PAT determination was carried out by DPV. There are also aptasensors developed by using the DPV technique for PAT determination in literature [19,22,23]. A PAT-specific aptamer was immobilized on the surface of EDC/NHS-activated electrode followed by the interaction of the aptamer and PAT on the electrode surface. DPV measurements were performed in the presence of a redox probe solution (i.e., 5 mM [Fe(CN)_6_]^3−/4−^). The analysis was performed on the change in the redox probe signal observed at +0.22 V. In the DPV, a peak observed in the specific potential (+0.22 V) occurs as a result of the reduction of [Fe(CN)_6_]^3−^ to [Fe(CN)_6_]^4−^ when the pulse is applied. As the reduction reaction takes place, a current is generated due to the flow of electrons between the working electrode and the redox couple in the solution and, accordingly, the resulting current is recorded. In this study, after the PGE was immersed into the PAT solution and the interaction of aptamer with PAT, a decrease at the current was recorded due to the formation of a barrier preventing electron transfer [18,35,36]. Determination of PAT was accordingly performed while measuring a decrease in current values that are proportional to the PAT concentration. The schematic representation of the developed voltammetric aptasensor is given in Figure 1.

### 3.1. Optimization of the Experimental Parameters for PAT Aptasensor

In this study, the experimental parameters, namely, aptamer concentration, aptamer immobilization time and interaction time, were optimized. First, the aptamer concentration was optimized. Different concentrations of PAT-specific aptamer, varying from 0.25 µg/mL to 5 µg/mL, were immobilized onto the electrode surface for 1 h after the EDC/NHS activation step. The interaction of 0.25 µg/mL PAT with aptamer was performed for 30 min and, accordingly, the voltammetric transduction was undertaken in a 5 mM redox probe solution using DPV (Appendix A). Due to the anionic nature of both the redox probe and aptamer sequence, a decrease at the current was recorded after aptamer immobilization onto the electrode surface as a result of the repulsive forces [35,36,37]. As the maximum decrease in current (33.62% decrease, *n* = 3) was obtained after immobilization of the 2.5 µg/mL aptamer onto the electrode’s surface (Appendix A), the optimum aptamer concentration was selected as 2.5 µg/mL.

Next, the effect of aptamer immobilization time upon the aptasensor’s response was studied. In this context, a 2.5 µg/mL aptamer was immobilized onto the electrode surface at different periods of time, 30 and 60 min. After 30 and 60 min of aptamer immobilization on the electrode surface, 33.17% and 33.62% decreases in current were observed in contrast with the control group, respectively. The electrode was then immersed into the solution of 0.25 µg/mL PAT and the interaction was performed at the electrode surface for 30 min. The % change at current was calculated according to the results observed before and after the interaction of aptamer with PAT (shown in Appendix A). By means of the interaction of the aptamer with its specific target, PAT at the electrode surface, a decrease occurred at the current due to the binding of PAT to its DNA aptamer, resulting in the formation of a barrier preventing electron transfer [18,35,36]. In the case of the 30 min aptamer immobilization time, and the respective aptamer interaction with PAT, no decrease at current was recorded as the aptamer may not be properly immobilized onto the surface over the 30 min. On the other hand, after 60 min of aptamer immobilization, a 29.41% decrease in current was recorded after its interaction with PAT (in Appendix A). Accordingly, the optimum aptamer immobilization time was chosen as 60 min. In the literature, there are also some aptasensor studies [38,39] that use the same aptamer immobilization time.

Finally, the effect of PAT interaction time upon the aptasensor response was explored. In this context, PAT interaction with aptamer was examined at various interaction times—15, 30 and 60 min—and the results are presented in Appendix A. When the interaction time was increased from 15 min to 30 min, a decrease in current was recorded as PAT interacts more efficiently with its DNA aptamer. However, a lower decrease ratio in current was recorded, possibly due to the saturation of the bio interaction of aptamer with PAT when the interaction time was increased from 30 min to 60 min. Because the highest decrease % in current was observed in the 30 min interaction time, this was chosen as the optimum interaction time for our further study. The experimental parameters studied and the selected values are summarized in Table 1.

### 3.2. Analytical Performance of PAT Aptasensor

Under optimum conditions, the analytical performance of a PAT-specific aptasensor was explored. The PAT solutions in concentrations varying from 1 pg/mL to 10^5^ pg/mL were prepared in PBS (50 mM, pH 7.40) and the aptasensor was then immersed into the PAT solutions for the interaction step. Finally, the DPV measurements were performed, and the results are shown in Appendix A. In the concentration range of PAT from 1 to 10^4^ pg/mL, a decrease in current was observed with an increase of the PAT concentration. The calibration plot was obtained based on the average current values (*n* = 3) measured in the PAT concentration range of 1–10^4^ pg/mL, with the following regression equation; y = −7.452 log(C_PAT_, pg/mL) + 112.17 (R^2^ = 0.9967) (Figure 2).

The limit of detection was calculated according to the IUPAC method [40], Sdl = Sreag + k ∗ σreag, (k = 3 [41] and σreag is the standard deviation of the blank with *n* = 3 parallel measurements) and found as 0.18 pg/mL (Appendix A).

In comparison with the earlier studies summarized in Table 2, it can be emphasized that the lower limit of detection was achieved in the present study for a wide range of PAT concentrations.

There are several studies, presented in Table 2, that present aptasensors developed for PAT determination [18,19,20,21,22,23,24,25,26,27,28,29]. Voltammetric analyses were mostly performed following research reviews on the electrochemical detection of PAT [19,22,23,24]. In all of these, the majority employed the DPV approach [19,22,23], though there are also studies using the EIS technique [18,25]. For example, in the study of He and Dong [22], a voltammetric aptasensor specific to PAT was developed while using a gold electrode modified with ZnO nanorods, with a chitosan composite being used for voltammetric determination, which was itself performed in a hexacyanoferrate redox probe solution. A lower limit of detection value of PAT was achieved in our study in comparison with the study of He and Dong [22] (shown in Table 2). In another study [18] an impedimetric aptasensor for PAT was reported by using a screen-printed carbon electrode modified with a carboxy–amine polyethylene glycol chain. Similarly, a lower limit of detection for PAT was obtained in our study comparison with that study [18]. In addition to electrochemical methods, there are fluorescence-based and SERS-based aptasensor studies developed for PAT analysis, and these are among the most frequently used techniques in the literature [20,21,27,28,29]. For example, Ahmadi et al. [27] developed a ratiometric fluorescence aptasensor for PAT. In that study, the ratiometric fluorescence response decreased in the presence of PAT due to disassembly of the DNA duplex structure and target-mediated release of the complementary DNA sequence. In contrast with the study of Ahmadi et al. [27], a lower limit of detection was obtained in our study (shown in Table 2). Guo et al. [21] have also reported a SERS-based aptasensor for PAT analysis. In the study of Guo et al. [21], an SERS aptasensor was developed by combining a gold–silver core–shell structure containing a signaling molecule and chitosan modified magnetic nanoparticles (CS-Fe_3_O_4_). In comparison with the LOD value obtained by the SERS aptasensor, a lower LOD was also achieved in our study. In addition, a lower LOD value was obtained in our study in contrast with the fluorescence, SERS and lateral flow assays presented in Table 2. Due to the advantages provided by electrochemical techniques, a more sensitive, selective and accurate aptasensor was established within our research. Taking into account the variety of methodologies used in previous research, a lower limit of detection was reached in our current work in contrast with these studies [18,20,21,22,24,26,27,28,29]. In addition, the ability to perform voltammetric analysis using a smartphone-connected aptasensor is important for bedside or on-site analysis. Moreover, the advantages of a voltammetric PAT aptasensor are found in their single-use application as well as their provision of a fast analysis with sensitive and selective results.

### 3.3. Selectivity of PAT Aptasensor

The selectivity of the aptasensor was investigated against ochratoxin A (OTA), deoxynivalenol (DON) and fumonisin B1 (FB1) mycotoxins. Amounts of 10 ng/mL of PAT and other mycotoxins—OTA, FB1 and DON—were prepared individually in PBS and DPV measurements were performed before and after the interaction process by following the same procedure. The results are shown in Figure 3. In the presence of unwanted interferents (OTA, DON, and FB1), no interaction effect was observed as no significant decrease in current was obtained. In comparison with the aptamer control group, a 31.57% decrease in current was recorded in the case of the aptamer interaction with PAT. When the electrodes were immersed into the individual solution of the other interferents—OTA, DON, and FB1—decreases in current of approximately 2.86%, 1.75%, and 1.96% were observed, respectively. Moreover, it was found that the data resembled the aptamer control signal. (Figure 3). As a PAT-specific aptamer sequence was used in the study, an aptamer–PAT interaction occurred at the electrode surface when the aptamer-immobilized electrodes were immersed in PAT solution. Conformational changes occurred after the aptamer–PAT interaction on the electrode surface created a barrier effect on electron transfer and a decrease in current was observed. When aptamer-immobilized electrodes were immersed in solutions of different mycotoxins, no interaction occurred and no significant decrease in current was observed. This decrease in current is in accordance with the studies in the literature [18,35,36]. In addition, a statistical test (called a *t*-test) was performed, in a similar manner to an earlier study [42] in order to see if the signals from each sample were different, assuming that both sets of samples had the same variability. The *p* values obtained from the *t*-test between PAT–OTA, PAT–DON and PAT–FB1 are 0.009, 0.006 and 0.013, respectively, and the *p* value for all groups is below 0.05. According to the results, it can be concluded that this aptasensor is very selective to PAT, due to the higher affinity of the DNA aptamer to its cognate mycotoxin PAT in contrast with other mycotoxins.

### 3.4. Application of Smartphone-Connected PAT Aptasensor in Apple Juice Samples

Since PAT is most commonly examined in apple juices [7,8,9], the implementation of the PAT aptasensor was demonstrated herein in these samples. Apple juice samples were purchased from the local market. Different preparation techniques for apple juice samples are outlined in the literature [19,22,43]. In general, these are laborious procedures that take a long time and involve many steps. Therefore, non-multi-step procedures are preferred. In this context, apple juice samples were diluted to a ratio of 1:50 using PBS. The PAT solutions in different concentrations (10^1^ to 10^5^ pg/mL) were prepared in diluted apple juice (1:50) and, accordingly, the voltammetric measurements were performed (Appendix A).

In order to present the further implementation of the PAT aptasensor within a portable device, PAT determination was performed using a smartphone-connected aptasensor. The PAT solutions, varying from 1 pg/mL to 10^4^ pg/mL, were prepared in 1:50 diluted apple juice. A decrease in current was obtained that was proportional with the increasing concentration of PAT in the sample. Based on these data, the calibration plot was obtained based on the average current values (*n* = 3) with the following regression equation: y = −13.473 log(C_PAT_, pg/mL) + 141.45 (R^2^ = 0.995) (Appendix A).

The limit of detection was calculated according to the IUPAC method [40], Sdl = Sreag + k ∗ σreag, (k = 3 [41] and σreag is the standard deviation of the blank with *n* = 3 parallel measurements) and was found to be 0.47 pg/mL in diluted apple juice (1:50) (Appendix A).

In addition, a picture of the set-up of the smartphone-integrated portable device is given in Appendix A.

### 3.5. Recovery and Reproducibility of PAT Aptasensor

The recovery study was carried out for PAT analysis in the samples of diluted apple juice by using the portable device with a smartphone-connected PAT aptasensor. Solutions of 1, 100, and 10,000 pg/mL PAT were added into diluted apple juice medium (1:50) and PAT determination was performed. The results are presented in Table 3. In the recovery study, low, medium and high concentrations were selected from the linear concentration range of PAT and the calculation was undertaken. As can be seen in Table 3, the recovery values obtained were between 91.24% and 93.47% at the three different concentration values of PAT, with at least three measurements. According to these results, it can be concluded that PAT determination can be performed in apple juice samples using a smartphone-connected aptasensor.

In addition, the reproducibility of the PAT aptasensor was examined using a smartphone-integrated portable device in apple juice medium. On three different days, measurements were carried out with two different aptasensors each day. The results are given in Appendix A. In the presence of 10,000 pg/mL PAT, the relative standard deviation (RSD) value was found to be 5.23% with six different aptasensors on different days (Appendix A). It can be concluded from the data that the developed aptasensor provides good repeatability and accurate results across batches.

## 4. Conclusions

In this study, a voltammetric aptasensor for PAT determination was performed and the application of a smartphone-connected PAT aptasensor was, for the first time, investigated in combination with a portable device. The advantages of this single-use aptasensor are found in its low cost and the selective, sensitive and accurate results it generates in a short time (the whole procedure, including analysis, is completed in 120 min). The limit of detection in buffer medium was estimated as 0.18 pg/mL in the range of 1–10^4^ pg/mL. The selectivity of the aptasensor was examined against DON, OTA and FB1 and it was found that the aptasensor recognized and detected PAT very selectively. In order to show the suitability of this aptasensor for further bedside or on-site analysis as a proof of concept, voltammetric determination of PAT was also performed using a smartphone-integrated portable device. To the best of our knowledge, this is the first aptasensor study in the literature that presents the voltammetric determination of PAT by a smartphone-integrated portable device. PAT determination was performed in apple juice medium using a smartphone-integrated portable device with an LOD of 0.47 pg/mL. In addition, a recovery range of PAT was obtained between 91.24% and 93.47%. The ability to perform analysis using a smartphone-connected aptasensor is also important for bedside analysis or on-site analysis. Nonetheless, there are numerous challenges and restrictions in the aptasensor development process. One of the most important challenges with electrochemical aptansensors is the need to ensure the reproducibility of the aptamer–target interaction. Nucleases and other environmental elements may cause aptamers to degrade, which could impair sensor function. Thus, other approaches to enhance aptamer stability are being explored, including chemical changes and immobilization procedures. Another point is that background noise from the sample matrix or non-specific binding can interfere with the detection of the target analyte, leading to false positive or false negative results. To increase the signal-to-noise ratio, sensor optimization is essential. The transition of electrochemical aptasensors from the laboratory to real-world applications can have additional challenges. For example, factors, such as sample preparation, interference from complex sample matrices, and the need for robust, user-friendly devices, all need to be considered to ensure the practicality and reliability of these sensors in the real world. In particular, on-site, fast and accurate analysis of food quality control is very important for both producers and consumers. It is predicted that such advanced technologies will become widespread in the future and will function not only in food security but also in environmental and health monitoring.

## Figures and Tables

**Figure 1 sensors-24-00754-f001:**
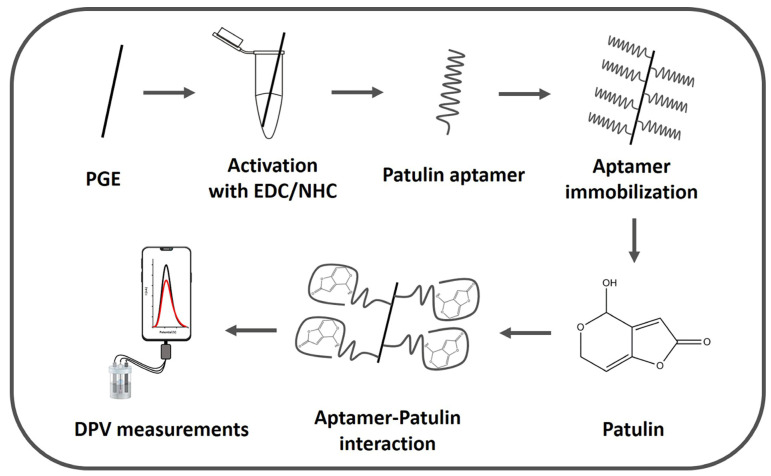
Schematic representation of the aptasensor for voltammetric detection of PAT by DPV. In the voltammograms, the black one represents the measurement obtained in the absence of PAT and the red one represents the measurement obtained in the presence of PAT.

**Figure 2 sensors-24-00754-f002:**
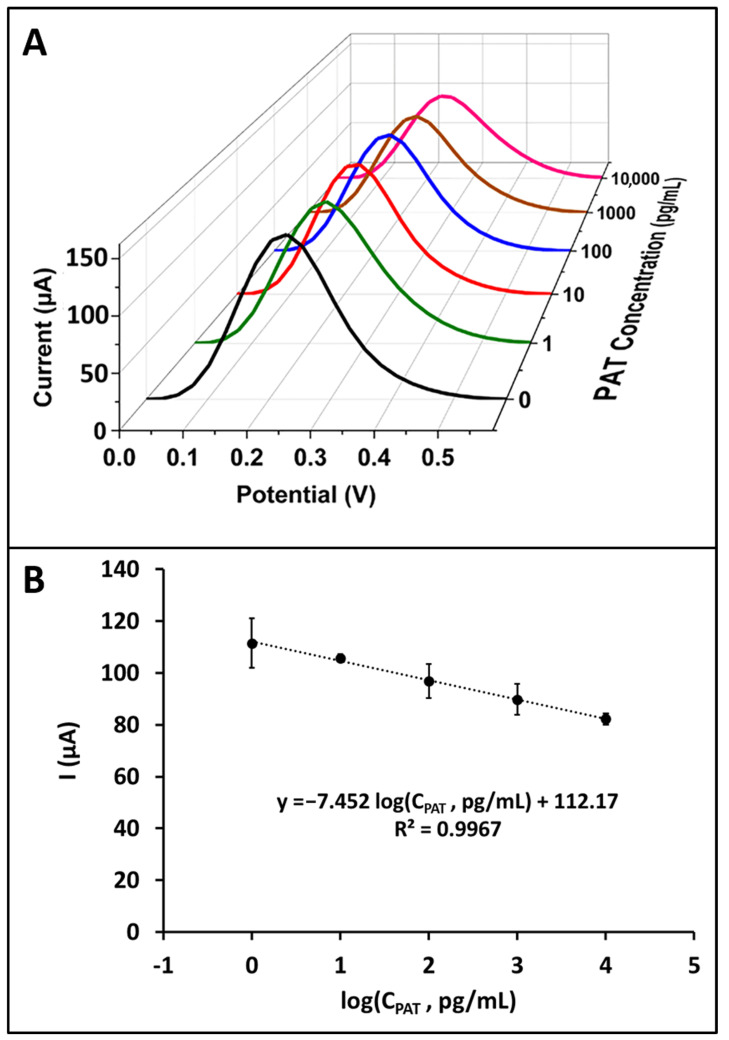
(**A**) Representative voltammograms presenting the redox probe signals observed in the concentration range of PAT from 1 pg/mL to 10^4^ pg/mL prepared in buffer medium. The voltammograms in black, green, red, blue, brown and pink colour represent 0 pg/mL (blank), 1 pg/mL, 10^1^ pg/mL, 10^2^ pg/mL, 10^3^ pg/mL and 10^4^ pg/mL PAT, respectively. (**B**) The resulting calibration plot presenting the average current values (*n* = 3) measured in the voltammetric determination of PAT.

**Figure 3 sensors-24-00754-f003:**
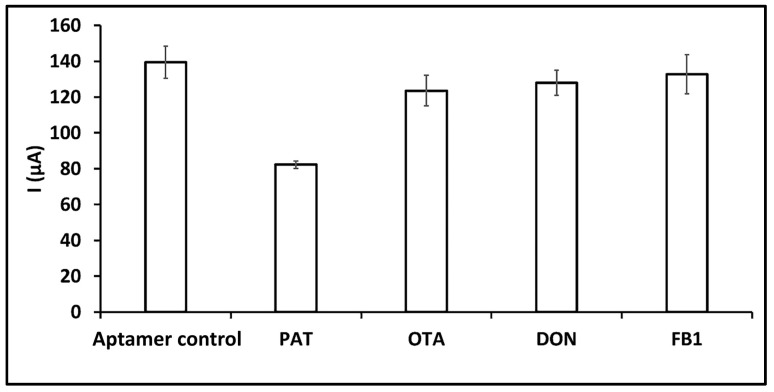
Histogram showing the average current values (*n* = 3) of the redox probe signal measured before and after interaction of aptamer with each of the mycotoxins in buffer medium.

**Table 1 sensors-24-00754-t001:** Optimization parameters and selected values.

Parameters	Tested Values	Selected Value
Aptamer concentration	0.25 µg/mL–1 µg/mL–2.5 µg/mL–5 µg/mL	2.5 µg/mL
Aptamer immobilization time	30 min and 60 min	60 min
PAT interaction time	15 min, 30 min, and 60 min	30 min

**Table 2 sensors-24-00754-t002:** Comparison of aptasensor studies developed for PAT analysis using different techniques.

Technique	Linear Range	LOD	Application	Smartphone Implementation	Reference
Electrochemical impedance spectroscopy	1–25 pg/mL	2.8 pg/mL	Apple juice samples	No	[18]
Differential pulse voltammetry	0.05 pg/mL–500,000 pg/mL	0.0304 pg/mL	Apple juice samples	No	[19]
Differential pulse voltammetry	0.5 pg/mL–50,000 pg/mL	0.27 pg/mL	Apple juice samples	No	[22]
Differential pulse voltammetry	0.05 pg/mL–500 pg/mL	0.0414 pg/mL	Apple juice and apple wine samples	No	[23]
Square wave voltammetry	0.1 nM–100 µM	0.043 nM	Apple, pear and tomato samples	No	[24]
Electrochemical impedance spectroscopy	1 nM–1 µM	0.3 µM	Apple juice samples	No	[25]
Lateral flow assay	2700–139,800 pg/mL in buffer and 7070–359,500 pg/mL in apple juice sample	190 pg/mL in buffer and 360 pg/mL apple juice sample	Apple juice samples	No	[26]
Fluorescent	15 pg/mL–35,000 pg/mL	6 pg/mL	Apple juice samples	No	[27]
Fluorescent	1 pg/mL–100,000 pg/mL	0.753 pg/mL	Apple juice samples	No	[28]
Fluorescent	20–1000 pg/mL in buffer and 50–1000 pg/mL apple juice sample	10 pg/mL in buffer and 30 pg/mL in apple juice sample	Apple juice samples	No	[29]
Fluorescent	10–10,000 pg/mL and 1000–200,000 pg/mL	7300 pg/mL	Apple juice samples	No	[20]
Surface-enhanced Raman spectroscopy	0–70,000 pg/mL	38.4 pg/mL	Apple samples	No	[21]
Differential pulse voltammetry	1–10,000 pg/mL in buffer and in apple juice medium	0.18 pg/mL in buffer and 0.47 pg/mL in apple juice medium	Apple juice samples	Yes	This work

**Table 3 sensors-24-00754-t003:** Recovery results obtained in diluted apple juice medium by using a smartphone-connected PAT aptasensor with portable device.

PAT Added (pg/mL)	PAT Found (pg/mL)	Recovery (%)	RSD (%)
1	0.92	92.00	9.22
100	93.47	93.47	5.11
10,000	9124.12	91.24	2.46

## Data Availability

Data are contained within the article.

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
