# Peer review of "Smartphone-Controlled Aptasensor for Voltammetric Detection of Patulin in Apple Juice"

_sensors, 2024, doi:10.3390/s24030754_

Round 1
Reviewer 1 Report
Comments and Suggestions for Authors
The manuscript titled "Smartphone Controlled Aptasensor for Voltammetric Detection of Patulin in Apple Juice" by Arzum Erdem and Huseyin Senturk presents a study on the development of a voltammetric aptasensor for detecting patulin (PAT), a mycotoxin, in apple juice. The study focuses on the design and evaluation of an aptasensor that integrates with a smartphone for ease of use and portability.
Here are some comments and suggestions for the manuscript:
1. The manuscript does not provide an adequate comparative analysis with existing methods for detecting patulin, which is essential for assessing the new aptasensor's relative performance and advancements.
2. The paper currently lacks a detailed discussion on the technological limitations and potential interference factors that could impact the sensor's performance in real-world applications.
3. A more robust statistical analysis of the sensor's performance data is needed. This should include comprehensive error analysis and confidence intervals to provide a clearer understanding of the sensor's accuracy and reliability.
Author Response
Manuscript ID: sensors-2792389
Type of manuscript: Article
Title: Smartphone controlled aptasensor for voltammetric detection of patulin in apple juice
Submitted to section: Biosensors,
https://www.mdpi.com/journal/sensors/sections/biosensors
Electrochemical DNA- and Aptasensors for the Detection of Low-Molecular
Compounds
https://www.mdpi.com/journal/sensors/special_issues/NUN8U0ET4S
December 27, 2023
The list of our answers to the comments of reviewers
Thank you for valuable comments of Editor, Reviewer 1, Reviewer 2, Reviewer 3, and Reviewer 4. We revised manuscript according to each comment pointed by editor and reviewers. The revised parts in the manuscript are highlighted in yellow.
Reviewer 1
Open Review
(x) I would not like to sign my review report
( ) I would like to sign my review report
Quality of English Language
(x) I am not qualified to assess the quality of English in this paper
( ) English very difficult to understand/incomprehensible
( ) Extensive editing of English language required
( ) Moderate editing of English language required
( ) Minor editing of English language required
( ) English language fine. No issues detected
|
Yes |
Can be improved |
Must be improved |
Not applicable |
|
|
Does the introduction provide sufficient background and include all relevant references? |
(x) |
( ) |
( ) |
( ) |
|
Are all the cited references relevant to the research? |
(x) |
( ) |
( ) |
( ) |
|
Is the research design appropriate? |
( ) |
(x) |
( ) |
( ) |
|
Are the methods adequately described? |
( ) |
(x) |
( ) |
( ) |
|
Are the results clearly presented? |
( ) |
(x) |
( ) |
( ) |
|
Are the conclusions supported by the results? |
(x) |
( ) |
( ) |
( ) |
Comments and Suggestions for Authors
The manuscript titled "Smartphone Controlled Aptasensor for Voltammetric Detection of Patulin in Apple Juice" by Arzum Erdem and Huseyin Senturk presents a study on the development of a voltammetric aptasensor for detecting patulin (PAT), a mycotoxin, in apple juice. The study focuses on the design and evaluation of an aptasensor that integrates with a smartphone for ease of use and portability.
Here are some comments and suggestions for the manuscript:
Answer: We would like to thank to Reviewer 1 for her/his valuable comments. We addressed all the issues and made the necessary changes one by one below according to his/her comments.
- The manuscript does not provide an adequate comparative analysis with existing methods for detecting patulin, which is essential for assessing the new aptasensor's relative performance and advancements.
Answer: Detailed comparisons with studies in the literature using conventional techniques have been added to the manuscript and are also given as follows:
“In comparison to earlier studies summarized in Table 2, it can be emphasized that the lower limit of detection was achieved in the present study in a wide range of PAT concentrations.
There are several studies presenting aptasensors developed for patulin determination [18–29] was shown in Table 2. Upon reviewing research on the electrochemical detection of patulin, the voltammetric analysis were mostly performed [19,22–24]. In all of them, the majority employed the DPV approach [19,22,23]. Apart from these, there are also studies using the EIS technique [18,25]. For example, in the study of He and Dong [22], a voltammetric aptasensor specific to patulin was developed while using the gold electrode modified with ZnO nanorods and chitosan composite was used for voltammetric determination performed in hexacyanoferrate redox probe solution. A lower limit value of PAT was achieved in our study in comparison to the study of He and Dong [22] (shown in Table 2). In another study [18] an impedimetric aptasensor for patulin was reported by using the screen-printed carbon electrode modified with carboxy-amine polyethylene glycol chain. Similarly, a lower limit of detection for PAT was recorded in our study comparison to that study [18]. In addition to electrochemical methods, there are fluorescence-based and SERS-based aptasensor studies developed for patulin analysis, which are among the frequently used techniques in the literature [20,21,27–29]. For example, Ahmadi et al. [27] developed a ratiometric fluorescence aptasensor for patulin. In the study, the ratiometric fluorescence response decreased in the presence of patulin due to disassembly of the DNA duplex structure and target-mediated release of the complementary DNA sequence. In contrast to the study of Ahmadi et al. [27], a lower limit of detection was obtained in our study (shown in Table 2). Guo et al. [21] also reported a SERS-based aptasensor for patulin analysis. In the study of Guo et al. [21], SERS aptensor was developed by combining a gold-silver core-shell structure containing a signalling molecule and chitosan modified magnetic nanoparticles (CS-Fe3O4). Comparison to the LOD value obtained by SERS aptasensor, a lower detection limit was also achieved in our study. In addition, the LOD value was obtained in our study in contrast to fluorescence, SERS and lateral flow assays presented in Table 2. Due to the advantages provided by electrochemical techniques, a more sensitive, selective and accurate aptasensor was established within our research. Taking into account the research using various methodologies, a lower limit of detection was reached in our current work in contrast to these studies [18,20–22,24,26–29]. In addition, the ability to perform voltammetric analysis using a smartphone connected aptasensor is also important furtherly for bedside analysis or on-site analysis. Moreover, the advantages of voltammetric PAT aptasensor are single-use application as well as providing fast analysis with the sensitive and selective results.”
Table 2. Comparison of aptasensor studies developed for PAT analysis by using different techniques
|
Technique |
Linear Range |
LOD |
Application |
Smartphone Implementation |
Reference |
|
Electrochemical impedance spectroscopy |
1 – 25 pg/mL |
2.8 pg/mL |
Apple juice samples |
No |
[18] |
|
Differential pulse voltammetry |
0.05 pg/mL – 500000 pg/mL |
0.0304 pg/mL |
Apple juice samples |
No |
[19] |
|
Differential pulse voltammetry |
0.5 pg/mL – 50000 pg/mL |
0.27 pg/mL |
Apple juice samples |
No |
[22] |
|
Differential pulse voltammetry |
0.05 pg/mL – 500 pg/mL |
0.0414 pg/mL |
Apple juice and apple wine samples |
No |
[23] |
|
Square wave voltammetry |
0.1 nM – 100 µM |
0.043 nM |
Apple, pear and tomato samples |
No |
[24] |
|
Electrochemical impedance spectroscopy |
1 nM – 1 µM |
0.3 µM |
Apple juice samples |
No |
[25] |
|
Lateral flow assay |
2700 – 139800 pg/mL in buffer and 7070 – 359500 pg/mL in apple juice sample |
190 pg/mL in buffer and 360 pg/mL apple juice sample |
Apple juice samples |
No |
[26] |
|
Fluorescent |
15 pg/mL – 35000 pg/mL |
6 pg/mL |
Apple juice samples |
No |
[27] |
|
Fluorescent |
1 pg/mL – 100000 pg/mL |
0.753 pg/mL |
Apple juice samples |
No |
[28] |
|
Fluorescent |
20 – 1000 pg/mL in buffer and 50 – 1000 pg/mL apple juice sample |
10 pg/mL in buffer and 30 pg/mL in apple juice sample |
Apple juice samples |
No |
[29] |
|
Fluorescent |
10 – 10000 pg/mL and 1000 – 200000 pg/mL |
7300 pg/mL |
Apple juice samples |
No |
[20] |
|
Surface-enhanced Raman spectroscopy |
0 – 70000 pg/mL |
38.4 pg/mL |
Apple samples |
No |
[21] |
|
Differential pulse voltammetry |
1 – 10000 pg/mL in buffer and in apple juice medium |
0.18 pg/mL in buffer and 0.47 pg/mL in apple juice medium |
Apple juice samples |
Yes |
This work |
- The paper currently lacks a detailed discussion on the technological limitations and potential interference factors that could impact the sensor's performance in real-world applications.
Answer: A detailed discussion of the limitations and potential challenges of aptasensors has been added to the manuscript and is also given as follows:
“The aptasensor developed in our work has the following benefits. Nonetheless, there are numerous challenges and restrictions in the aptasensor development process. One of the most important challenges with electrochemical aptansensors is to ensure the reproducibility of the aptamer-target interaction. Nucleases and other environmental elements may cause aptamers to degrade, which could impair sensor function. Thus, other approaches to enhance aptamer stability are being explored, including chemical changes and immobilization procedures. Another point is that background noise from the sample matrix or non-specific binding can interfere with the detection of the target analyte, leading to false positive or false negative results. To increase the signal-to-noise ratio, sensor optimization is essential. The transition of electrochemical aptasensors from the laboratory to real-world applications can have additional challenges. For example, factors such as sample preparation, interference from complex sample matrices, and the need for robust, user-friendly devices all need to be considered to ensure the practicality and reliability of these sensors in the real world.”
- A more robust statistical analysis of the sensor's performance data is needed. This should include comprehensive error analysis and confidence intervals to provide a clearer understanding of the sensor's accuracy and reliability.
Answer: In the selectivity studies (3.3. Selectivity of PAT aptasensor), statistical test (t-test) was performed. t-test was used to calculate the p Values and it was found that the p Values were below 0.05. These results confirmed that the developed aptasensor showed high selectivity to PAT. In the literature, there are also studies in which statistical analyses are used in a simialar way [41]. The relevant selectivity studies section has been added to the manuscript and is also given as follows:
“3.3. Selectivity of PAT aptasensor
The selectivity of the aptasensor was investigated against to ochratoxin A (OTA), deoxynivalenol (DON) and fumonisin B1 (FB1) mycotoxins. 10 ng/mL of PAT and other mycotoxins; OTA, FB1 and DON were prepared individually in PBS and DPV measurements were performed before and after interaction process by following the same procedure. The results were shown in Figure 3. In the presence of unwanted substituents (OTA, DON, and FB1), no interaction effect was observed since no significant decrease in current was obtained. In comparison to the aptamer control group, a 31.57% decrease in current was recorded in the case of the aptamer interaction with patulin. When the electrodes immersed into the individual solution of other interferents; OTA, DON, and FB1, a decrease in current about the ratio of 2.86%, 1.75%, and 1.96% was observed respectively. Moreover, it was found that the data resembled the aptamer control signal. (Figure 3). Since a patulin-specific aptamer sequence was used in the study, aptamer-patulin interaction occurred at the electrode surface when the aptamer immobilised electrodes were immersed in patulin solution. Conformational changes occurring after aptamer-patulin interaction on the electrode surface created a barrier effect on electron transfer and a decrease in current was observed. When aptamer immobilised electrodes were immersed in solutions of different mycotoxins, no interaction occurred and no significant decrease in current was observed. This decrease in current is in accordance with the studies in the literature [18,35,36]. In addition, a statistical test (called a t-test) was performed similarly to earlier study [41] in order to see if the signals from each sample were different, assuming that both sets of samples had the same variability. The p values obtained from the t-test between PAT-OTA, PAT-DON and PAT-FB1 are 0.009, 0.006 and 0.013, respectively, and the p value for all groups is below 0.05. According to the results, it could be concluded that this aptasensor is very selective to patulin due to the higher affinity of DNA aptamer to its cognate mycotoxin PAT in contrast to other mycotoxins.”
Figure 3. Histogram showing the average current values (n=3) of redox probe signal measured before and after interaction of aptamer with each of mycotoxins in buffer medium.

Reviewer 2 Report
Comments and Suggestions for Authors
The manuscript by Erdem et al. reported the development and application of a portable electrochemical sensors for patulin detection in apple juice. The integration with a cell phone makes it potentially powerful portable device. I have a few comments and suggestions listed below.
1. The immobilization time of aptamer was tested with 30 and 60 min. It is necessary to include at least 3 different time for optimization.
2. The reproducibility among different batches of device should be evaluated to demonstrate its feasibility in a apple juice analysis.
3. There are significant differences in current values between Fig.2 and Fig.3. What makes them so different?
4. It would be nice to include a picture of the device.
5. Pay attention to the large space in Line 186
Author Response
Manuscript ID: sensors-2792389
Type of manuscript: Article
Title: Smartphone controlled aptasensor for voltammetric detection of patulin in apple juice
Submitted to section: Biosensors,
https://www.mdpi.com/journal/sensors/sections/biosensors
Electrochemical DNA- and Aptasensors for the Detection of Low-Molecular
Compounds
https://www.mdpi.com/journal/sensors/special_issues/NUN8U0ET4S
December 27, 2023
The list of our answers to the comments of reviewers
Thank you for valuable comments of Editor, Reviewer 1, Reviewer 2, Reviewer 3, and Reviewer 4. We revised manuscript according to each comment pointed by editor and reviewers. The revised parts in the manuscript are highlighted in yellow.
Reviewer 2
Open Review
(x) I would not like to sign my review report
( ) I would like to sign my review report
Quality of English Language
( ) I am not qualified to assess the quality of English in this paper
( ) English very difficult to understand/incomprehensible
( ) Extensive editing of English language required
( ) Moderate editing of English language required
( ) Minor editing of English language required
(x) English language fine. No issues detected
|
Yes |
Can be improved |
Must be improved |
Not applicable |
|
|
Does the introduction provide sufficient background and include all relevant references? |
(x) |
( ) |
( ) |
( ) |
|
Are all the cited references relevant to the research? |
(x) |
( ) |
( ) |
( ) |
|
Is the research design appropriate? |
( ) |
(x) |
( ) |
( ) |
|
Are the methods adequately described? |
(x) |
( ) |
( ) |
( ) |
|
Are the results clearly presented? |
( ) |
(x) |
( ) |
( ) |
|
Are the conclusions supported by the results? |
(x) |
( ) |
( ) |
( ) |
Comments and Suggestions for Authors
The manuscript by Erdem et al. reported the development and application of a portable electrochemical sensors for patulin detection in apple juice. The integration with a cell phone makes it potentially powerful portable device. I have a few comments and suggestions listed below.
Answer: We would like to thank to Reviewer 2 for her/his valuable comments. We addressed all the issues and made the necessary changes one by one below according to his/her comments.
- The immobilization time of aptamer was tested with 30 and 60 min. It is necessary to include at least 3 different time for optimization.
Answer: These immobilization times were selected and tested for optimization study based on the knowledge from previous aptasensor studies. In the optimization of aptamer immobilization time, similar current values (142.07 ± 2.10 µA, n=3 for 30 min aptamer immobilization time and 141.11 ± 6.30 µA, n=3 for 60 min aptamer immobilization time) were recorded both 30 min and 60 min aptamer immobilization onto the electrode surface and a similar decrease (33.17% decrease at 30 min and 33.62% decrease at 60 min) in current was observed in comparison to the result of the control group. As a result, it was decided that no need to examine any other immobilization time. In the literature, there are also some aptasensor studies [38,39] performed same aptamer immobilisation time. The relevant section has been included into the manuscript and also given as follows:
“Next, the effect of aptamer immobilization time upon to aptasensor’s response was studied. In this context, 2.5 µg/mL aptamer was immobilized onto the electrode surface in different period of time; 30 and 60 min. After 30 min and 60 min aptamer immobilization on the electrode surface, 33.17% and 33.62% decrease in current was observed in contrast to the control group, respectively. The electrode was then immersed into the solution of 0.25 µg/mL PAT and interaction was performed at the electrode surface for 30 min. The % change at current was calculated according to the results observed before and after the interaction of aptamer with patulin (shown in Table S2). By means of the interaction of the aptamer with its specific target, PAT at the electrode surface, it was resulted with a decrease at the current due to binding of PAT to its DNA aptamer resulted with the formation of a barrier preventing electron transfer [18,35,36]. In the case of 30 min aptamer immobilization time, and accordingly aptamer interaction with PAT, no decrease at current was recorded since the aptamer may not be properly immobilized onto the surface during 30 min. On the other hand, 60 min aptamer immobilization time, a 29.41% decrease in current was recorded after its interaction with PAT (in Table S2). Accordingly, the optimum aptamer immobilization time was chosen as 60 min. In the literature, there are also some aptasensor studies [38,39] performed same aptamer immobilisation time.”
- The reproducibility among different batches of device should be evaluated to demonstrate its feasibility in a apple juice analysis.
Answer: The reproducibility of the developed aptasensor against different batches was investigated in apple juice medium. The relevant section has been revised. The revised section is given in the manuscript, Supplementary Material as follows:
In the manuscript:
“In addition, the reproducibility of PAT aptasensor was examined using a smartphone integrated to portable device in apple juice medium. On three different days, measurements were carried out with two different aptasensors each day. The results are given in Table S6. In the presence of 10000 pg/mL PAT, the relative standard deviation (RSD) value was found to be 5.23% with 6 different aptasensors on different days (Table S6). It can be concluded from the data that the developed aptasensor has provided a good repeatability and provides accurate results across batches.
In the Supplementary Material:
Table S6. Reproducibility of PAT aptasensor using a smartphone integrated to portable device in apple juice medium in three different days.
|
10000 pg/mL PAT |
I (µA) |
RSD (%, n=6) |
|
1st day |
85.81 |
5.23 |
|
84.61 |
||
|
2st day |
78.25 |
|
|
89.44 |
||
|
3th day |
78.79 |
|
|
85.67 |
- There are significant differences in current values between Fig.2 and Fig.3. What makes them so different?
Answer: In the selectivity study, a batch of experiments was carried out and the relevant section was revised. According to the results shown in Fig. 2 and Fig. 3, the consistent results were obtained. Furthermore, the discussion was detailed in the selectivity study section, in addition, statistical test (t-test) was performed. The revised section has been added to the manuscript and is given below.
“3.3. Selectivity of PAT aptasensor
The selectivity of the aptasensor was investigated against to ochratoxin A (OTA), deoxynivalenol (DON) and fumonisin B1 (FB1) mycotoxins. 10 ng/mL of PAT and other mycotoxins; OTA, FB1 and DON were prepared individually in PBS and DPV measurements were performed before and after interaction process by following the same procedure. The results were shown in Figure 3. In the presence of unwanted substituents (OTA, DON, and FB1), no interaction effect was observed since no significant decrease in current was obtained. In comparison to the aptamer control group, a 31.57% decrease in current was recorded in the case of the aptamer interaction with patulin. When the electrodes immersed into the individual solution of other interferents; OTA, DON, and FB1, a decrease in current about the ratio of 2.86%, 1.75%, and 1.96% was observed respectively. Moreover, it was found that the data resembled the aptamer control signal. (Figure 3). Since a patulin-specific aptamer sequence was used in the study, aptamer-patulin interaction occurred at the electrode surface when the aptamer immobilised electrodes were immersed in patulin solution. Conformational changes occurring after aptamer-patulin interaction on the electrode surface created a barrier effect on electron transfer and a decrease in current was observed. When aptamer immobilised electrodes were immersed in solutions of different mycotoxins, no interaction occurred and no significant decrease in current was observed. This decrease in current is in accordance with the studies in the literature [18,35,36]. In addition, a statistical test (called a t-test) was performed similarly to earlier study [41] in order to see if the signals from each sample were different, assuming that both sets of samples had the same variability. The p values obtained from the t-test between PAT-OTA, PAT-DON and PAT-FB1 are 0.009, 0.006 and 0.013, respectively, and the p value for all groups is below 0.05. According to the results, it could be concluded that this aptasensor is very selective to patulin due to the higher affinity of DNA aptamer to its cognate mycotoxin PAT in contrast to other mycotoxins.
Figure 3. Histogram showing the average current values (n=3) of redox probe signal measured before and after interaction of aptamer with each of mycotoxins in buffer medium.
- It would be nice to include a picture of the device.
Answer: The picture of the set-up of the smartphone integrated portable device is added to the Supporting Material and is also given below.
Figure S4. Picture of the set-up of the smartphone integrated portable device.
- Pay attention to the large space in Line 186
Answer: The relevant line has been checked and corrected. Furthermore, all manuscript was similarly checked.

Reviewer 3 Report
Comments and Suggestions for Authors
The task that the authors have chosen for analysis is very relevant. It is relevant because there is a wide distribution of this type of product – apples, apple products – juices in the first place. Food consumption is very high. At the same time, it is safe to say that there is practically no diagnosis for the presence of PATULIN. And even more so, an operational diagnosis based on immunodetection (ID).
The authors have chosen an interesting solution. To combine immunodetection with modern diagnostic methods – namely, the use of enzymatic analysis in combination with modern ID methods.
I will not dwell on the approach used - the pairing of reaction and testing. He deserves high praise.
In my opinion, the use of APTAMERS for virus detection deserves the highest praise. This is a really new approach. A new approach based precisely on the use of this type of APTAMER for this type of Antigen – meaning PATULIN.
Satisfactory results have been obtained.
The limit of detection (LOD) was found to be 0.18 pg/mL in the range of 1 – 104 pg/mL of Patulin in buffer medium under optimum experimental conditions.
The selectivity of PAT aptasensor against to ochratoxin A, fumonisin B1 and deoxynivalenol mycotoxins was examined and it was found that the aptasensor was very selective to patulin.
The LOD was achieved as 0.47 pg/mL in diluted apple juice medium. In addition, a recovery range of 91.24% - 93.47% was obtained for patulin detection.
The study was carried out using modern methodology. The accuracy of the results is beyond doubt.
The results should be of interest to readers. The manuscript can be accepted for printing without edits.
Author Response
Manuscript ID: sensors-2792389
Type of manuscript: Article
Title: Smartphone controlled aptasensor for voltammetric detection of patulin in apple juice
Submitted to section: Biosensors,
https://www.mdpi.com/journal/sensors/sections/biosensors
Electrochemical DNA- and Aptasensors for the Detection of Low-Molecular
Compounds
https://www.mdpi.com/journal/sensors/special_issues/NUN8U0ET4S
December 27, 2023
The list of our answers to the comments of reviewers
Thank you for valuable comments of Editor, Reviewer 1, Reviewer 2, Reviewer 3, and Reviewer 4. We revised manuscript according to each comment pointed by editor and reviewers. The revised parts in the manuscript are highlighted in yellow.
Reviewer 3
Open Review
( ) I would not like to sign my review report
(x) I would like to sign my review report
Quality of English Language
( ) I am not qualified to assess the quality of English in this paper
( ) English very difficult to understand/incomprehensible
( ) Extensive editing of English language required
( ) Moderate editing of English language required
( ) Minor editing of English language required
(x) English language fine. No issues detected
|
Yes |
Can be improved |
Must be improved |
Not applicable |
|
|
Does the introduction provide sufficient background and include all relevant references? |
(x) |
( ) |
( ) |
( ) |
|
Are all the cited references relevant to the research? |
(x) |
( ) |
( ) |
( ) |
|
Is the research design appropriate? |
(x) |
( ) |
( ) |
( ) |
|
Are the methods adequately described? |
(x) |
( ) |
( ) |
( ) |
|
Are the results clearly presented? |
(x) |
( ) |
( ) |
( ) |
|
Are the conclusions supported by the results? |
(x) |
( ) |
( ) |
( ) |
Comments and Suggestions for Authors
The task that the authors have chosen for analysis is very relevant. It is relevant because there is a wide distribution of this type of product – apples, apple products – juices in the first place. Food consumption is very high. At the same time, it is safe to say that there is practically no diagnosis for the presence of PATULIN. And even more so, an operational diagnosis based on immunodetection (ID).
The authors have chosen an interesting solution. To combine immunodetection with modern diagnostic methods – namely, the use of enzymatic analysis in combination with modern ID methods.
I will not dwell on the approach used - the pairing of reaction and testing. He deserves high praise.
In my opinion, the use of APTAMERS for virus detection deserves the highest praise. This is a really new approach. A new approach based precisely on the use of this type of APTAMER for this type of Antigen – meaning PATULIN.
Satisfactory results have been obtained.
The limit of detection (LOD) was found to be 0.18 pg/mL in the range of 1 – 104 pg/mL of Patulin in buffer medium under optimum experimental conditions.
The selectivity of PAT aptasensor against to ochratoxin A, fumonisin B1 and deoxynivalenol mycotoxins was examined and it was found that the aptasensor was very selective to patulin.
The LOD was achieved as 0.47 pg/mL in diluted apple juice medium. In addition, a recovery range of 91.24% - 93.47% was obtained for patulin detection.
The study was carried out using modern methodology. The accuracy of the results is beyond doubt.
The results should be of interest to readers. The manuscript can be accepted for printing without edits.
Answer: We would like to thank very much to Reviewer 3 for her/his valuable comments on our manuscript.

Reviewer 4 Report
Comments and Suggestions for Authors
This work presented an electrochemical sensor for detecting patulin in apple juice. The sensor showed excellent LOD and high selectivity among a variety of other substances. The manuscript is well written and organized. I would suggest accepting it after addressing the following comments.
1. It would be great if the author could introduce more about the current research status of electrochemical sensors in the introduction section. For example, what specific techniques have been used for the detection.
2. Would the authors please explain why DPV technique was chosen for the detection?
3. The author should discuss the electrochemical mechanisms of the DPV results. For example, what kind of redox reaction occurred for the peak?
Author Response
Manuscript ID: sensors-2792389
Type of manuscript: Article
Title: Smartphone controlled aptasensor for voltammetric detection of patulin in apple juice
Submitted to section: Biosensors,
https://www.mdpi.com/journal/sensors/sections/biosensors
Electrochemical DNA- and Aptasensors for the Detection of Low-Molecular
Compounds
https://www.mdpi.com/journal/sensors/special_issues/NUN8U0ET4S
December 27, 2023
The list of our answers to the comments of reviewers
Thank you for valuable comments of Editor, Reviewer 1, Reviewer 2, Reviewer 3, and Reviewer 4. We revised manuscript according to each comment pointed by editor and reviewers. The revised parts in the manuscript are highlighted in yellow.
Reviewer 4
Open Review
(x) I would not like to sign my review report
( ) I would like to sign my review report
Quality of English Language
( ) I am not qualified to assess the quality of English in this paper
( ) English very difficult to understand/incomprehensible
( ) Extensive editing of English language required
( ) Moderate editing of English language required
( ) Minor editing of English language required
(x) English language fine. No issues detected
|
Yes |
Can be improved |
Must be improved |
Not applicable |
|
|
Does the introduction provide sufficient background and include all relevant references? |
(x) |
( ) |
( ) |
( ) |
|
Are all the cited references relevant to the research? |
(x) |
( ) |
( ) |
( ) |
|
Is the research design appropriate? |
(x) |
( ) |
( ) |
( ) |
|
Are the methods adequately described? |
(x) |
( ) |
( ) |
( ) |
|
Are the results clearly presented? |
(x) |
( ) |
( ) |
( ) |
|
Are the conclusions supported by the results? |
(x) |
( ) |
( ) |
( ) |
Comments and Suggestions for Authors
This work presented an electrochemical sensor for detecting patulin in apple juice. The sensor showed excellent LOD and high selectivity among a variety of other substances. The manuscript is well written and organized. I would suggest accepting it after addressing the following comments.
Answer: We would like to thank to Reviewer 4 for her/his valuable comments. We addressed all the issues and made the necessary changes one by one below according to his/her comments.
- It would be great if the author could introduce more about the current research status of electrochemical sensors in the introduction section. For example, what specific techniques have been used for the detection.
Answer: As mentioned by the reviewer, detailed information about electrochemical sensors is given in the Introduction section. The revised section are given in the manuscript as follows:
“There are several studies presenting aptasensors specific to PAT [18–29]. In these studies, the biosensors based on electrochemical [18,19,22–25], optical [20,27–29] and SERS-based [21] methods have been introduced. In recent years, electrochemical methods have emerged as a promising alternative for detecting different analytes. Recent years have seen a great deal of research and development on electrochemical sensors, with a focus on improving their resilience, usefulness, and appropriateness for new sensing goals. The number of various nanomaterials used in sensor design has risen, particularly with the advancement of nanotechnology, and the developed sensors' performance has significantly improved. These nanomaterials can provide a large surface area and offer unique electronic properties that improve sensor performance. Furthermore, it is possible to detect biological analytes such proteins and different nucleic acids with a high degree of specificity by integrating biological recognition components, such as enzymes, antibodies, DNA, aptamer etc. with electrochemical sensors. Miniaturized electrochemical sensor development is becoming more popular these days for applications such as continuous monitoring and point-of-care diagnostics. These sensors are appropriate for environmental monitoring and personalized healthcare because of their low cost, portability, and real-time measurement capabilities. In biosensor studies, a variety of electrochemical techniques are employed to appraise and analyze the performance of sensors for various applications. Among these techniques, cyclic voltammetry, differential pulse voltammetry, square wave voltammetry, amperometry and electrochemical impedance spectroscopy techniques are widely used. The use of electrochemical techniques offers several potential advantages for example, the advantages of electrochemical techniques are high sensitivity, selective analysis, low detection limits, fast analysis, suitable for miniaturization, and relatively inexpensive [30,31].”
- Would the authors please explain why DPV technique was chosen for the detection?
Answer: Information about why the DPV technique was chosen in the study has been added to the manuscript. The revised section is indicated in the manuscript and also added below.
“In this study, a voltammetric transduction was employed for the determination of PAT. Differential pulse voltammetry (DPV) is a highly sensitive electrochemical technique widely used in biosensor studies because of the many benefits it provides. The most important advantages are its sensitivity, selectivity, enabling measurements over a wide potential range, low background current and fast measurement. For these reasons, patulin determination was carried out by DPV technique. There are also aptasensors developed by using DPV technique for patulin determination in literature [19,22,23].”
- The author should discuss the electrochemical mechanisms of the DPV results. For example, what kind of redox reaction occurred for the peak?
Answer: The electrochemical mechanism of the DPV results is added to the manuscript as follows:
……… “DPV measurements were performed in the presence of redox probe solution (i.e, 5 mM [Fe(CN)6]3-/4- ). The analysis were performed on the change in the redox probe signal observed at +0.22 V. In the DPV technique, a peak observed in the specific potential (+0.22 V) occurs as a result of the reduction of [Fe(CN)6]3- to [Fe(CN)6]4- when the pulse is applied. As the reduction reaction takes place, a current is generated due to the flow of electrons between the working electrode and the redox couple in the solution and accordingly, the resulting current is recorded. In this study, after the PGE was immersed into the PAT solution and the interaction of aptamer with PAT, a decrease at the current was recorded due to the formation of a barrier preventing electron transfer [18,35,36]. Quantitative analysis of PAT was accordingly performed while measuring a decrease at current values that are proportional to the PAT concentration.” ………
In addition, the reason for the increase and decrease in current in the presence of a redox probe was already discussed in the previous version of our manuscript. The relevant parts are also shown below:
……. “Due to the anionic nature of both the redox probe and aptamer sequence, a decrease at current was recorded after aptamer immobilization onto electrode surface as a result of the repulsive forces [35–37].” ……..
…….. “By means of the interaction of the aptamer with its specific target, PAT at the electrode surface, it was resulted with a decrease at the current due to binding of PAT to its DNA aptamer resulted with the formation of a barrier preventing electron transfer [18,35,36].” ………

Round 2
Reviewer 1 Report
Comments and Suggestions for Authors
The MS was revised carefully by the authors and can be accepted.
Author Response
Comments and Suggestions for Authors
The MS was revised carefully by the authors and can be accepted.
Answer: Thank you very much for the referee's valuable comments.